# Advantageous Strain Sensing Performances of FBG Strain Sensors Equipped with Planar UV-Curable Resin

**DOI:** 10.3390/s23052811

**Published:** 2023-03-03

**Authors:** Xiaojin Li, Min Zheng, Dan Hou, Qiao Wen

**Affiliations:** 1Key Laboratory of Optoelectronic Devices and Systems of Ministry of Education and Guangdong Province, College of Physics and Optoelectronic Engineering, Shenzhen University, Shenzhen 518060, China; 2Optical Fiber Sensing Engineering Technology R&D Center of Guangdong Province, T&S Communications Co., Ltd., Shenzhen 518118, China

**Keywords:** FBG strain sensors, planar UV-curable resin, excellent strain sensing properties

## Abstract

The existing optical strain sensors based on fiber Bragg grating (FBG) have limitations, such as a complex structure, a limited strain range (±200 με) and poor linearity performance (R-squared value < 0.9920); these limitations affect their potential practical applications. Here, four FBG strain sensors equipped with planar UV-curable resin are investigated. The proposed FBG strain sensors have a simple structure, a large strain range (±1800 με) and excellent linearity performance (R-squared value ≥ 0.9998); they further produce the following performances: (1) good optical properties, including an undistorted Bragg peak shape, narrow bandwidth (−3 dB bandwidth ≤ 0.65 nm) and a high side mode suppression ratio (SMSR, the absolute value of SMSR ≥ 15 dB); (2) good temperature sensing properties with high temperature sensitivities (≥47.7 pm/°C) and a good linearity performance (R-squared value ≥ 0.9990); and (3) excellent strain sensing properties with no hysteresis behavior (hysteresis error ≤ 0.058%) and excellent repeatability (repeatability error ≤ 0.045%). Based on their excellent properties, the proposed FBG strain sensors are expected to be applied as high-performance strain sensing devices.

## 1. Introduction

Surface strain measurement is one of the most important monitoring methods for the structural health monitoring of aerospace, civil and mechanical engineering buildings and infrastructures [1,2]. As a measuring tool, electric strain gauge (ESG) sensors are widely used to monitor surface strain [1,2,3,4]. However, in some harsh environments, such as those with high temperatures, high radiation and high pressures [5], ESG sensors are not suitable for strain monitoring because of their disadvantages of non-insulation, susceptibility to electromagnetic interference and a short lifetime [6]. However, fiber Bragg grating (FBG) sensors have received extensive attention and are very suitable for harsh environment applications due to possessing many excellent advantages, such as electrical isolation, multiplexing, long-service life and immunity to electromagnetic interference [7,8]. Typically, FBG sensors are often used to measure temperature [9,10] and strain [11,12]. However, an important limitation of the engineering applications of FBG sensors is the sensitivity of the wavelength to both temperature and strain [13]. To address this problem, Nascimento et al. used the reference FBG method to discriminate the temperature and bidirectional strain [14]. Mokhtar et al. enhanced the discrimination between the temperature and strain in real sensing applications by designing a sensor packaging structure [15]. In addition, a hybrid sensing network (constituting of an FBG and Fabry-Perot cavity) was proposed to discriminate the temperature and strain in 2019 [16]. Moreover, Huang et al. decoupled temperature and stress by adjusting fiber morphologies [17]. As a common FBG sensor, the FBG strain sensors are often used to measure the deformation of a structure [18,19]. Typically, uncoated FBG strain sensors have a strain sensitivity of 1.2 pm/με for a Bragg wavelength of 1550 nm. To enhance the strain sensitivity of FBG strain sensors, a sensitivity-enhanced FBG strain sensor based on a substrate with a lever structure was developed by Li et al. [20]. In addition, a higher sensitivity FBG strain sensor based on a flexible hinge bridge displacement magnification structure was designed by Liu et al. [21]. However, the structural complexity of the abovementioned FBG strain sensors is a real challenge. In addition, the strain range (±200 με) of the abovementioned FBG strain sensors is limited because of the use of an uncoated FBG with low mechanical strength. However, material-coated (such as metal-coated or polymer-coated) FBG sensors have high mechanical strength [22,23]. Recently, a novel Fabry-Perot interferometer (FPI) fabrication method based on ultraviolet (UV)-resin coating was proposed [24]. The linear relationship curves of the wavelength and the strain for FPI based on UV- curable resin were obtained; however, the linearity performance of the FPI was not sufficient (i.e., R-squared value < 0.9990) [25]. Moreover, the linearity performance of the UV-resin coated FBG sensors was even lower (i.e., R-squared value < 0.9920) [26]. In addition, resin coated composite samples with polarimetric fiber or highly birefringent fiber sensors embedded in the multi-layer composite structure are fabricated [27,28], but the strain linearity of the polarimetric fiber sensors (R-squared value ≤ 0.9992), the hysteresis behavior and the repetition of the sensors were limited. The abovementioned limitations, i.e., the complex structure, limited strain range and poor linearity performance, affect their potential practical applications.

In this study, four FBG strain sensors equipped with planar UV-curable resin are fabricated. The proposed FBG strain sensors have a simple structure, a large strain range (±1800 με) and an excellent linearity performance (R-squared value ≥ 0.9998). In addition, the hysteresis behavior and the repeatability of the four FBG strain sensors are discussed. The experimental results show that the four FBG strain sensors have excellent strain sensing properties, including no hysteresis behavior (hysteresis error ≤ 0.058%) and excellent repeatability (repeatability error ≤ 0.045%). Because of these advantages, the proposed FBG strain sensors are anticipated to be applied as high-performance strain sensing devices.

## 2. Principle and Fabrication of the FBG Strain Sensor

### 2.1. Principle of the Uncoated FBG and Coated FBG

The FBG is a kind of periodic refractive index modulation structure along with the fiber core and is fabricated by exposing the photosensitized fiber core to ultraviolet (UV) light. The structure and principle of the uncoated FBG are illustrated in Figure 1a. If incident/broadband light is directed into the FBG, it will reflect a narrowband spectral component (i.e., the reflected light), called the Bragg wavelength ($\lambda_{B}$). According to Bragg’s law, the Bragg wavelength is expressed as follows [21]:(1)$$\lambda_{B}=2n_{eff}\cdot \Lambda$$
where $n_{eff}$ is the effective refractive index of the fiber core and Λ is the grating period. The Bragg wavelength will be shifted when the FBG is exposed to an external physical perturbation, such as heat and stress. The Bragg wavelength shift (Δλ) induced by ambient temperature change (ΔT) and the axial strain acting on the FBG (ε) is expressed as follows [21]:(2)$$\Delta \lambda /\lambda_{B}=(1-P_{e})\cdot \varepsilon +(\alpha +\xi )\cdot \Delta T$$
where $P_{e}$, α and ξ are the photo-elastic constant, thermal expansion coefficient and thermo-optic coefficient of the optical fiber, respectively. If the ambient temperature does not change, Equation (2) will be expressed as follows:(3)$$\Delta \lambda /\lambda_{B}=(1-P_{e})\cdot \varepsilon$$
typically, the strain sensitivity of an uncoated FBG is approximately 1.2 pm/με when $\lambda_{B}=1550\;\mathrm{nm}$ and $P_{e}=0.22$. However, for a material coated FBG, the strain sensitivity has a slight difference due to the presence of the coated material. Figure 1b shows the structure and left view of the FBG coated with a planar UV-curable resin. The planar UV-curable resin requires UV-irradiation and heating treatment. As shown in Figure 1b, the FBG is completely embedded into the planar UV-curable resin.

### 2.2. Fabrication of Uncoated FBG Strain Sensors

First, the pristine SMF-28e optical fibers (coated with acrylate) were obtained from Corning. Then, the high-pressure hydrogen loading process (to improve the photosensitivity of optical fibers) was carried out. After, the mechanical stripping (to remove the original acrylate coating) was needed. The stripping length was approximately 5 mm (and the grating length was 3 mm) as shown in Figure 1a. The fabrication of the uncoated FBG strain sensors is shown in Figure 2. Initially, a UV laser beam, emitted from the 248 nm excimer laser, passed through the beam shaping assembly (constituted by a cylindrical lens and aperture) and vertically entered the phase mask. After entering the phase mask, it was divided into three main parts (i.e., +1 order, −1 order and 0 order diffraction). The interference fringes (to write the FBG) formed by both +1 order and −1 order diffraction occurred in the vicinity of the stripped optical fiber. The FBG was written after the interference fringes acted on the optical fiber core. Moreover, an amplified spontaneous emission (ASE) broadband light source emitted the broadband light and it was injected into the abovementioned FBG. An optical spectrum analyzer was used to monitor the FBG spectrum in real-time. The optical fiber was fixed by two fiber clamps. Finally, the annealing process was needed to remove the excessive hydrogen and enhance the stability of the FBG. After finishing the above steps, high-quality (i.e., possessing an undistorted Bragg peak shape, −3 dB bandwidth ≤ 0.65 nm and absolute value of side mode suppression ratio (SMSR) ≥ 15 dB) FBG optical fibers were produced and selected as the uncoated (or initial) FBG strain sensors.

### 2.3. Fabrication of the FBG Strain Sensors

After preparing the uncoated FBG strain sensors, the fabrication process of the FBG strain sensors equipped with planar UV-curable resin was carried out as shown in Figure 3a,b where the schematic diagrams and the real photos are shown, respectively. First, a stainless steel substrate with a size of 7^(x)^ × 19^(y)^ × 0.06^(z)^ mm^3^ was prepared and then two limited blocks with a size of 2^(x)^ × 19^(y)^ × 0.60^(z)^ mm^3^ were attached to the substrate by Kapton tape. Subsequently, the uncoated FBG sensor was fixed onto the substrate by the Kapton tape. Third, the UV-curable resin (Isitic-3410) with a viscosity ≥14,200 mPa·s was injected into the groove formed by the two blocks and the substrate. After the injection, to remain on the same plane as the surfaces of the two limited blocks, the excess UV-curable resin was removed and its surface was smoothed by using a blade. Fourth, the injected UV-curable resin was exposed to UV irradiation at 395 nm (light intensity ≥ 100 mW/cm^2^) for 10 s and then heated at 90 °C for one hour. Finally, the Kapton tape was removed and the fabrication process of the FBG strain sensors was finished. Here, four FBG strain sensors (i.e., FBG-S1, FBG-S2, FBG-S3 and FBG-S4) with different thicknesses of UV-curable resin were successfully fabricated. Notably, the fabrication of the FBG strain sensors was very simple, without any complex structures.

## 3. Results and Discussion

### 3.1. Optical Properties of the FBG Strain Sensors

To understand the influence of the planar UV-curable resin on the uncoated FBG strain sensors, a comparison of the optical parameters of the FBG strain sensors before coating (uncoated or initial state) and after coating (i.e., after UV irradiation and heating) have been carried out, as listed in Table 1. After coating, the central wavelengths of the four FBG strain sensors (i.e., FBG-S1, FBG-S2, FBG-S3 and FBG-S4) decreased. This decrease is attributed to the compression of the grating period induced by the thermal expansion and contraction of the planar UV-curable resin. Since the planar UV-curable resin (with a high coefficient of thermal expansion) is completely cured at 90 °C and is then dropped to 25 °C, it tends to shrink and causes a compression of the grating due to the tight bonding force between the planar UV-curable resin and the grating. However, the changes in the −3 dB bandwidth and SMSR do not show any regularity. Additionally, they meet the requirement of high-quality FBG strain sensors (i.e., −3 dB bandwidth ≤ 0.65 nm and the absolute value of SMSR ≥ 15 dB). The SMSR of the FBG-S1 decreased (−20.50 dB → −15.25 dB) after coating due to the non-uniform strain distribution along the grating; when the grating is subjected to the non-uniform strain [29,30], the intensity of the side lobe (i.e., at 1538.919 nm in Figure 4a) increases asymmetrically with an increasing strain gradient. As the non-uniform strain gradient increases, the spectrum is gradually broadened. The increase (0.512 nm → 0.517 nm) in the −3 dB bandwidth is proven. According to the previous work [23], the non-uniform strain is caused by the asymmetric axial force induced by the asymmetric coating. In addition, the reflection intensity of the FBG-S1 is reduced after coating. According to the reflection intensity expression [31], a larger full width at half maximum leads to a smaller reflection intensity. Thus, the increase in the −3 dB bandwidth is the reason for the reduction in intensity.

In addition, spectral comparisons of the FBG strain sensors before and after coating have been also carried out as shown in Figure 4. After UV irradiation and heating treatment, the Bragg peaks of the four FBG strain sensors (i.e., FBG-S1, FBG-S2, FBG-S3 and FBG-S4) are not distorted when compared with the initial state. Specifically, the four FBG strain sensors have good optical properties with an undistorted Bragg peak shape, a narrow bandwidth and a high SMSR. Note that FBG-S3 shown in Figure 4c possesses the largest change in wavelength (i.e., 1.264 nm).

Based on previous experience in our laboratory, the difference in wavelength change is associated with the coating thickness; the thicker coating possesses a stronger bonding force and makes it easier to pull the grating, resulting in a more apparent change in wavelength. To better illustrate this, the relationship between the shift of the Bragg wavelength and the coating thickness is obtained as shown in Figure 5. A thicker UV-curable resin coating leads to a larger shift in the wavelength; the major difference among the four FBG strain sensors is that they have different coating thicknesses.

### 3.2. Temperature Sensing Properties of the FBG Strain Sensors

In addition to studying the optical properties, the temperature sensing properties of the FBG strain sensors were investigated as shown in Figure 6. Good linear relationships between the wavelength (y) and temperature (t) are exhibited in the whole measured temperature range. The linear relationships for FBG-S1, FBG-S2, FBG-S3 and FBG-S4 are y = 0.0527t + 1536.9787 (the temperature sensitivity is 52.7 pm/°C), y = 0.0477t + 1537.6557 (the temperature sensitivity is 47.7 pm/°C), y = 0.0557t + 1536.8882 (the temperature sensitivity is 55.7 pm/°C) and y = 0.0556t + 1537.0886 (the temperature sensitivity is 55.6 pm/°C), respectively. All four FBG strain sensors have high temperature sensitivity, i.e., ~5 times larger than the temperature sensitivities of the uncoated FBG strain sensors. Notably, due to possessing the thickest UV-curable resin, FBG-S3 has the highest temperature sensitivity of 55.7 pm/°C. Moreover, the coefficients of determination (i.e., R-squared value) for FBG-S1, FBG-S2, FBG-S3 and FBG-S4 are 0.9991, 0.9992, 0.9992 and 0.9990, respectively; the four FBG strain sensors have good linearity (R-squared value ≥ 0.9990). According to our engineering experience, the R-squared value ≥0.9990 further indicates that the four FBG strain sensors have good package properties. In summary, the four proposed FBG strain sensors have excellent temperature sensing properties with high temperature sensitivity and good linearity.

In addition, according to the previous work [32], the temperature sensitivity of the FBG embedded in the coating material is expressed as follows:(4)$$\Delta \lambda /\Delta T=\lambda_{B}\cdot \left[\xi +(1-P_{e})\cdot \alpha_{m}\right]\cdot \frac{C_{m}\cdot E_{m}}{C_{m}\cdot E_{m}+C_{f}\cdot E_{f}}$$
where $\alpha_{m}$ is the thermal expansion coefficient of the coating material. $C_{m}$ and $C_{f}$ are the cross-section areas of the coating material and the silica fiber. $E_{m}$ and $E_{f}$ are the Young’s modulus values of the coating material and the silica fiber. In our experiments, the coating material is the planar UV-curable resin. $\alpha_{m}=8.5\times 10^{-5}/{}^{\circ}C$ and $E_{m}=500\;\mathrm{MPa}$ are obtained from the product specification of the UV-curable resin. $C_{m}$ is equal to the product of the width and the thickness of the coating because the shape of the UV-curable resin is planar. Thus, one can see that the thicker coating leads to a larger $C_{m}$, resulting in a larger temperature sensitivity. For the silica fiber, $\xi =8.6\times 10^{-6}/{}^{\circ}C$ and $E_{f}=72\;\mathrm{GPa}$ [32,33]. $C_{f}=\pi r^{2}$, r is the radius (i.e., 62.5 μm) of the silica fiber. Thus, according to Equation (4), the temperature sensitivities of the four FBG strain sensors can be calculated; the calculated temperature sensitivities are 55.6 pm/°C, 53.9 pm/°C, 60.1 pm/°C and 56.7 pm/°C for FBG-S1, FBG-S2, FBG-S3 and FBG-S4, respectively. However, the temperature sensitivities obtained from the experiments (as shown in Figure 6) are 52.7 pm/°C, 47.7 pm/°C, 55.7 pm/°C and 55.6 pm/°C for FBG-S1, FBG-S2, FBG-S3 and FBG-S4, respectively. Specifically, there are some differences between the calculated sensitivities and the experimentally measured sensitivities probably because the experimentally measured sensitivities are influenced and limited by the stainless steel substrate. The temperature sensitivities of the various FBG sensors reported in the last 20 years [34,35,36,37,38,39,40,41,42,43,44,45,46,47] are listed in Table 2. To better compare with the other FBG sensors, the maximum measured sensitivity (i.e., 55.7 pm/°C) is used in Table 2. The second highest temperature sensitivity was achieved by our sensor from our study; this result produces a competitive advantage for our proposed FBG strain sensors when they are considered as FBG temperature sensors before installation. However, after installation, the strain sensing properties of the proposed FBG strain sensors further improved.

### 3.3. Strain Sensing Properties of the FBG Strain Sensors

To study the strain sensing properties of the FBG strain sensors, an experimental setup based on the equal strength cantilever beam was designed as shown in Figure 7; it includes four parts, i.e., the equal strength cantilever beam device, strain gauge, laptop and FBG wavelength demodulator (T&S Communication Co, Ltd., Shenzhen, China, TS-WI) with a wavelength resolution of 1 pm). Four ESG sensors (i.e., ESG-S1, ESG-S2, ESG-S3 and ESG-S4) and four FBG strain sensors (i.e., FBG-S1, FBG-S2, FBG-S3 and FBG-S4) are mounted on the upper and lower sides of the equal strength cantilever beam (the material is high manganese steel). The sensors attached to the upper side of the cantilever beam are used to measure the tensile strains and the sensors attached to the lower side of the cantilever beam are used to measure the compressive strains. Six weights are prepared to add an applied force from 0 N to 27 N. As shown in Figure 7 the temperature sensor based on the FBG is used on the cantilever beam device to record the ambient temperature change. The linear relationship for the temperature sensor is y = 0.0100 (t − 25.0) + 1532.9240 (the temperature sensitivity is 10.0 pm/°C). The strain gauge is used to demodulate the strain signals of the four ESG sensors and the FBG wavelength demodulator is used to demodulate the wavelength signals of the four FBG strain sensors. The experimental data and spectra are recorded by the laptop. To eliminate the influence of temperature, the experimental temperature is maintained at 21.0 °C and monitored by the abovementioned FBG temperature sensor during the whole process of the strain measurement.

According to material mechanics, for the strain sensors pasted on the equal strength cantilever beam, the absolute value of the strain (ε) can be expressed as follows [3,48]:(5)$$\varepsilon =\frac{6F\cdot L}{W\cdot h^{2}\cdot E}$$
where F is the applied force exerted on the cantilever beam. L is the distance from the sensor to the point with the added weight. W is the width of the point where the sensor pasted. E and h are the elasticity modulus and thickness of the cantilever beam, respectively. For our experiments, E=210 GPa and h=2 mm.

Initially, the relationship curves between the strain and the applied force for the four ESG sensors were obtained as shown in Figure 8. According to Equation (5) and Table 3, the theoretical values can be calculated and the measured values can be directly obtained from the strain gauge; the measured strains are in good agreement with the theoretical strains in the measured force range. Specifically, the measured values are nearly equal to the theoretical values in the range of 0–22 N. Notably, ESG-S1 and ESG-S3 have a positive strain value due to their pasting on the upper side of the cantilever beam. ESG-S2 and ESG-S4 have a negative strain value because of their attachment on the opposite side of the cantilever beam.

However, for FBG strain sensors, the measured strains cannot be directly obtained from the FBG wavelength demodulator because it is only used to demodulate wavelength signals. According to the results shown in Figure 8, the measured values are approximately equal to the theoretical values in the measured force range. Based on this result, the measured strains for the FBG strain sensors can be indirectly obtained from the practical values measured by the ESG sensors according to Equation (4) and the installation information listed in Table 3 as shown in Figure 9. The indirectly obtained strains are in good agreement with the theoretical strains in the measured force range. In the range of 0–22 N, the indirectly obtained values are nearly equal to the theoretical values. FBG-S1 and FBG-S3 have a positive strain value because of their pasting on the upper side of the cantilever beam. FBG-S2 and FBG-S4 have a negative strain value due to their attachment on the opposite side of the cantilever beam.

Although the relationship curves between the strain and the applied force for the FBG strain sensors were obtained, they could not directly reflect the relationship between the wavelength shift and the strain. To determine this, the relationship curves between the wavelength and the strain are obtained as shown in Figure 10. The four FBG strain sensors exhibit good linear relationships between the wavelength (y) and the strain (x) in the whole measured strain range. The strain sensitivities (i.e., ~1.5 pm/με) of the four FBG strain sensors are approximately equal to those values of the uncoated FBG sensors (i.e., ~1.2 pm/με). The coating thickness appears to have a slight influence on the strain sensitivity. Additionally, the R-squared values for FBG-S1, FBG-S2, FBG-S3 and FBG-S4 are 0.9998, 0.9998, 1.0000 and 1.0000, respectively. This means that the four FBG strain sensors have an excellent linearity performance (R-squared value ≥ 0.9998).

We also have studied the hysteresis behavior of the four FBG strain sensors during the loading and unloading processes obtained by adding and removing weights. The measured results are shown in Figure 11. The four FBG strain sensors exhibit an excellent linear fitting and an excellent linearity performance (R-squared value ≥ 0.9998). The hysteresis behavior cannot be observed in the whole measured strain range. Notably, FBG-S2 has a maximum difference (0.023 nm) at ~566 με. Thus, the hysteresis error (~0.058%), i.e., the ratio of the maximum difference to the full scale of the sensor (i.e., 40 nm for our FBG sensors), can be obtained according to the previous work [49] and this error is accepted for most practical applications.

Finally, we studied the repeatability of the four FBG strain sensors during three repeated loading processes and the measured results are shown in Figure 12. The four FBG strain sensors exhibit excellent linear fitting and repeatability. The FBG-S1 has a maximum difference (0.018 nm) at ~1747 με. According to the previous work [49], a repeatability error (~0.045%) can be obtained and this error is also acceptable.

## 4. Conclusions

In summary, we have investigated the optical properties, temperature sensing properties and strain sensing properties of four FBG strain sensors equipped with a planar UV-curable resin. The experimental results show that the proposed FBG strain sensors provided the following: (1) good optical properties, including an undistorted Bragg peak shape, narrow bandwidth and high SMSR; (2) good temperature sensing properties, including high temperature sensitivities (≥47.7 pm/°C) and good linearity (R-squared value ≥ 0.9990); and (3) excellent strain sensing properties, including excellent linearity performance (R-squared value ≥ 0.9998), no hysteresis behavior (hysteresis error ≤ 0.058%) and excellent repeatability (repeatability error ≤ 0.045%). In addition, when compared with the uncoated FBG sensors, the coating thickness of the UV-curable resin has an influence on the wavelength shifts and temperature sensitivities of the FBG strain sensors after coating. The proposed FBG strain sensors are expected to be applied as high-performance strain sensing devices due to their abovementioned good performances.

## Figures and Tables

**Figure 1 sensors-23-02811-f001:**
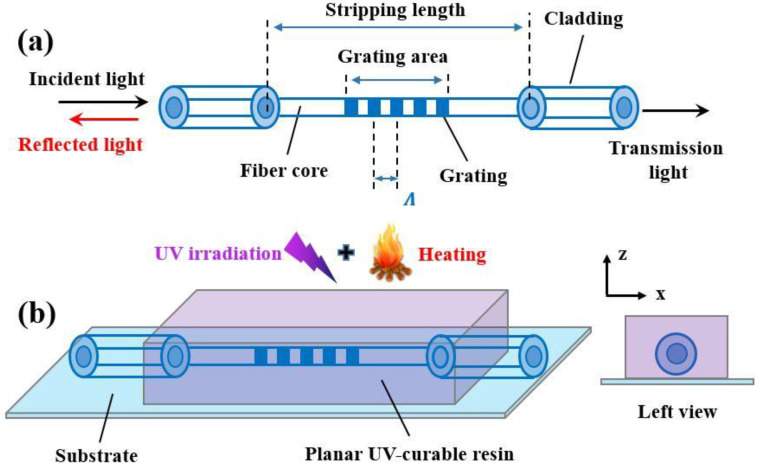
(**a**) Structure and principle of the uncoated FBG and (**b**) structure and left view of the FBG coated with a planar UV-curable resin. The substrate material is 304 stainless steel.

**Figure 2 sensors-23-02811-f002:**
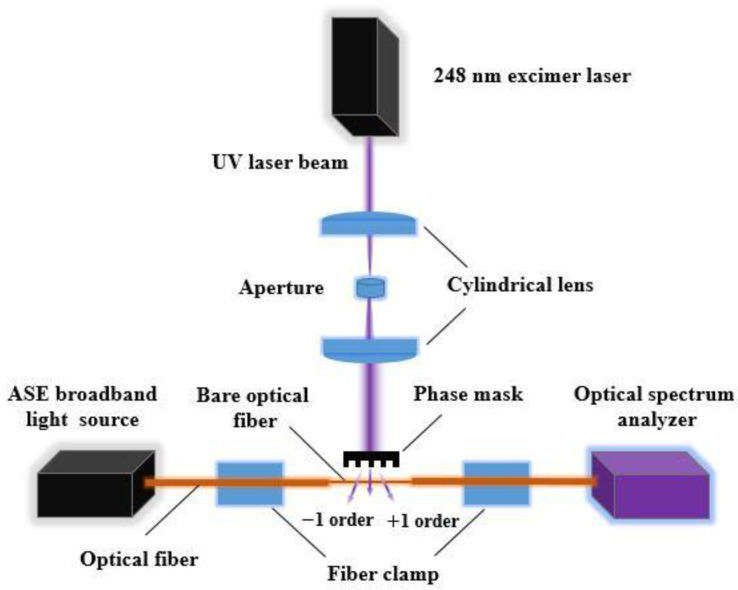
Schematic diagram of the experimental setup for fabricating the uncoated FBG sensors using UV laser based phase mask technology.

**Figure 3 sensors-23-02811-f003:**
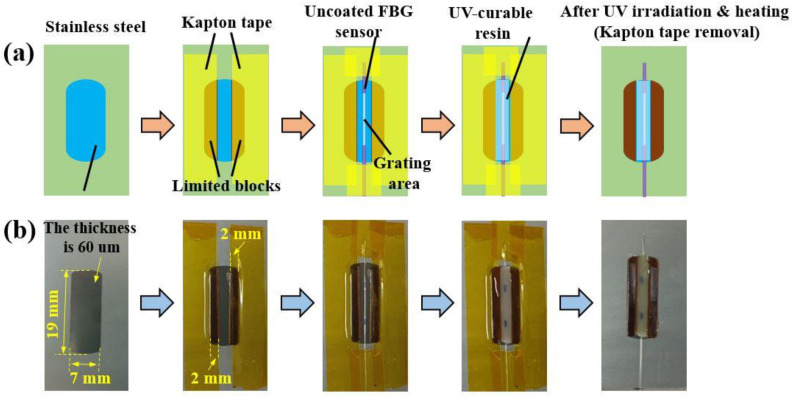
The process for fabricating the FBG strain sensors. (**a**) Schematic diagrams and (**b**) real photos, same as schematic diagrams.

**Figure 4 sensors-23-02811-f004:**
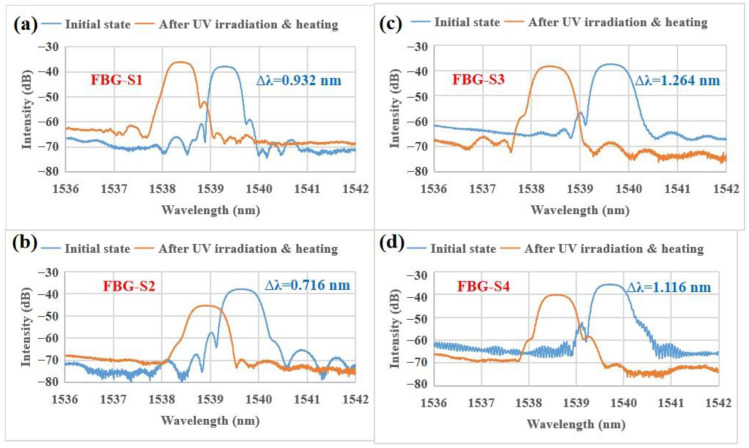
Spectral comparisons of the FBG strain sensors before coating and after UV irradiation and heating: (**a**) FBG-S1, (**b**) FBG-S2, (**c**) FBG-S3, (**d**) FBG-S4. The optical spectra were measured by an optical spectrum analyzer (Yokogawa AQ6370C) and the test temperature was 25.0 °C.

**Figure 5 sensors-23-02811-f005:**
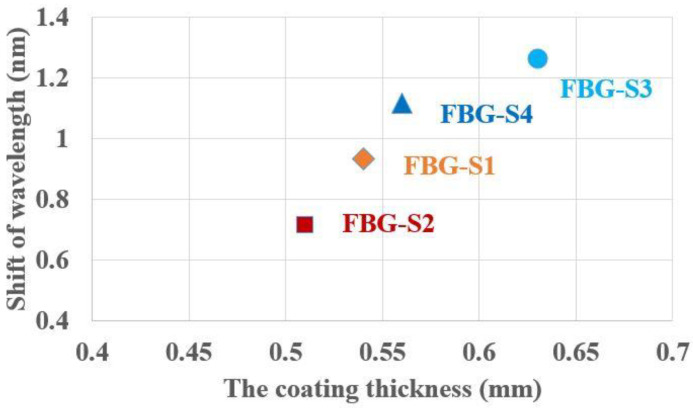
The relationship between the shift of the Bragg wavelength and the coating thickness (the coating thicknesses of the UV-curable resin for FBG-S1, FBG-S2, FBG-S3 and FBG-S4 are 0.54 mm, 0.51 mm, 0.63 mm and 0.56 mm, respectively).

**Figure 6 sensors-23-02811-f006:**
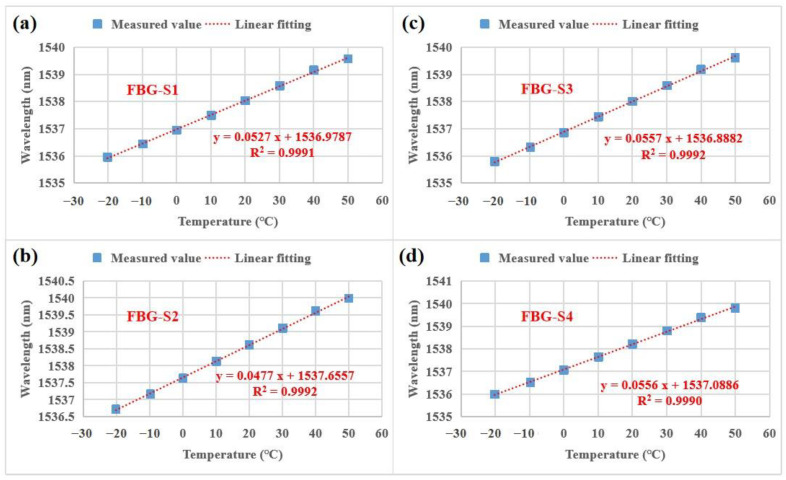
Central wavelengths of the FBG strain sensors as a function of temperature from −20 °C to 50 °C: (**a**) FBG-S1, (**b**) FBG-S2, (**c**) FBG-S3, (**d**) FBG-S4. The temperatures were program-controlled with a high and low temperature/humidity test chamber.

**Figure 7 sensors-23-02811-f007:**
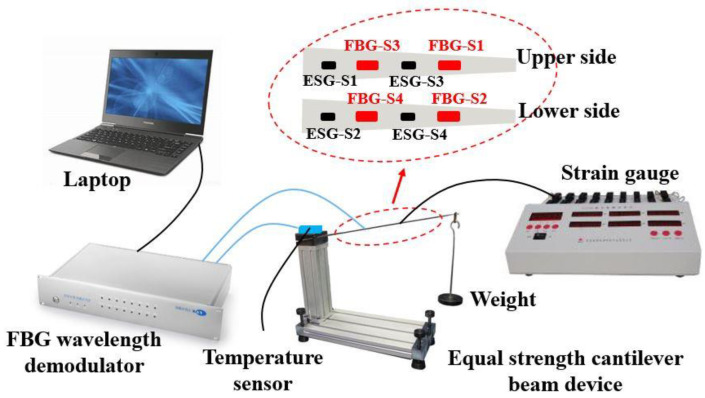
Schematic of the experimental setup for strain measurements.

**Figure 8 sensors-23-02811-f008:**
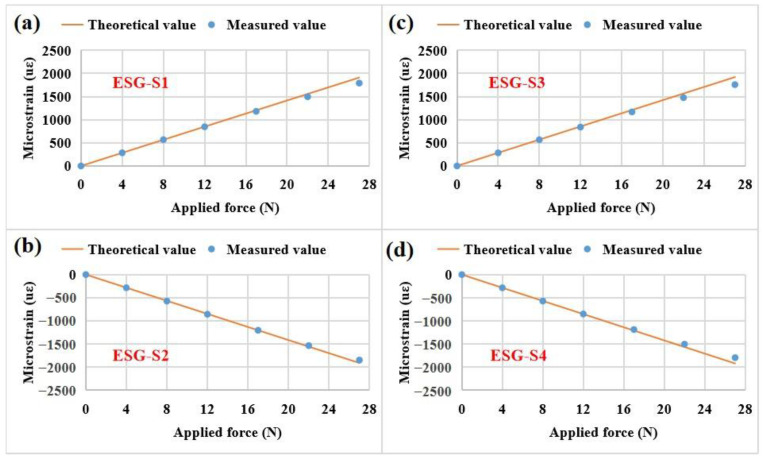
Comparisons of the theoretical and practical microstrains of the four ESG sensors mounted on the cantilever beam as a function of the applied forces from 0 N to 27 N: (**a**) ESG-S1, (**b**) ESG-S2, (**c**) ESG-S3, (**d**) ESG-S4.

**Figure 9 sensors-23-02811-f009:**
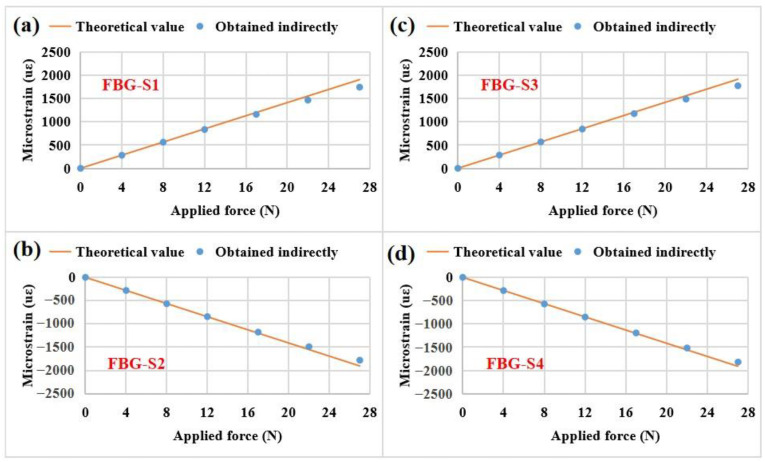
Comparisons of the theoretical and practical microstrains of the four FBG strain sensors mounted on the cantilever beam as a function of the applied forces from 0 N to 27 N: (**a**) FBG-S1, (**b**) FBG-S2, (**c**) FBG-S3, (**d**) FBG-S4.

**Figure 10 sensors-23-02811-f010:**
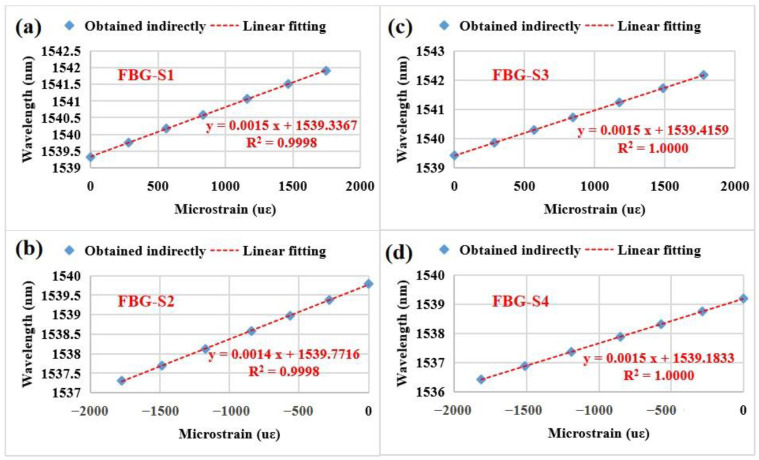
Central wavelengths of the four FBG strain sensors as a function of microstrains: (**a**) FBG-S1, (**b**) FBG-S2, (**c**) FBG-S3, (**d**) FBG-S4.

**Figure 11 sensors-23-02811-f011:**
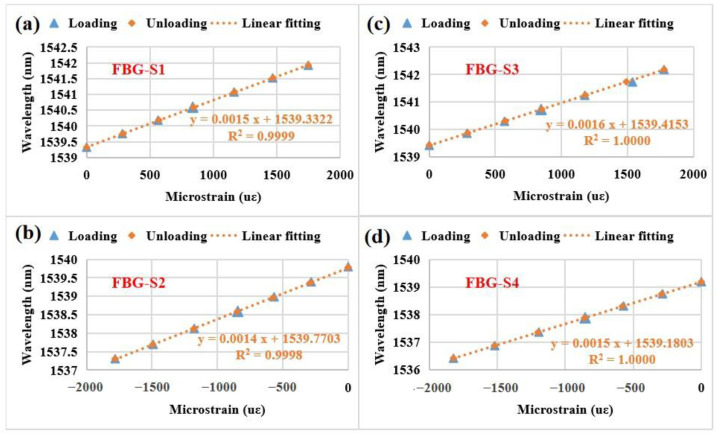
Central wavelengths of the four FBG strain sensors as a function of microstrains during the loading and unloading processes: (**a**) FBG-S1, (**b**) FBG-S2, (**c**) FBG-S3, (**d**) FBG-S4.

**Figure 12 sensors-23-02811-f012:**
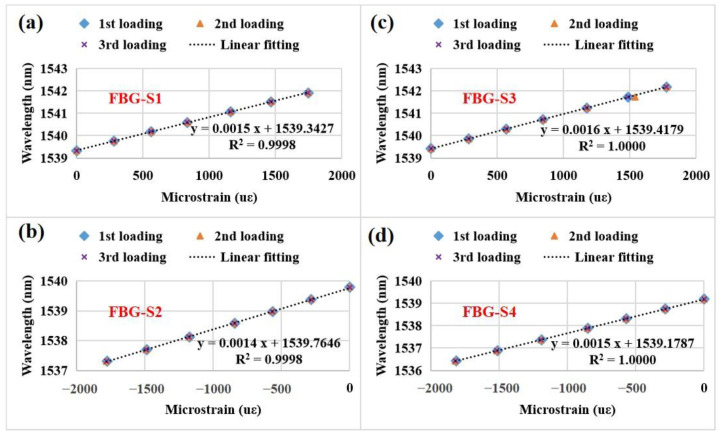
Central wavelengths of the four FBG strain sensors as a function of microstrains during three repeated loading processes: (**a**) FBG-S1, (**b**) FBG-S2, (**c**) FBG-S3, (**d**) FBG-S4.

**Table 1 sensors-23-02811-t001:** Optical parameter comparison of the FBG strain sensors in the initial state and after UV irradiation and heating. The test temperature was 25.0 °C.

	FBG-S1		FBG-S2		FBG-S3		FBG-S4	
State	Initial State	Coated State	Initial State	Coated State	Initial State	Coated State	Initial State	Coated State
Wavelength/nm	1539.299	1538.367	1539.636	1538.920	1539.623	1538.359	1539.675	1538.559
−3 dB bandwidth/nm	0.512	0.517	0.632	0.630	0.634	0.632	0.618	0.612
SMSR/dB	−20.50	−15.25	−19.50	−23.75	−18.87	−20.75	−17.12	−17.25

**Table 2 sensors-23-02811-t002:** Performance comparisons of various FBG sensors reported in the last 20 years (in chronological order).

Year	Materials/Design	Measuring Temperature (°C)		Maximum Sensitivity (pm/°C)	Reference
		Minimum	Maximum		
2001	FBG fixed on Teflon substrate	−196	27	150	[34]
2004	Strong FBG	25	950	15.14	[35]
2008	Metal coated FBG	−269	27	33.5	[36]
2011	Teflon coated FBG	−196	25	12.85	[37]
2013	An embedded FBG	30	90	30.9	[38]
2016	Chromium nitride coated FBG	100	650	14	[39]
2018	Polymer coated FBG	−180	25	48	[40]
2019	Fs-FBG	300	1000	15.9	[41]
2020	FBG based on vortex light	27	427	14.42	[42]
2020	FBG embedded diaphragm	20	35	49.8	[43]
2021	Titanium nitride coated FBG	−195	20	10.71	[44]
2021	Normal FBG	26	50	8.75	[45]
2022	Metallic packaged FBG	5	50	28.9	[46]
2022	Integrated FBG	0	250	40	[47]
2023	Planar UV-curable resin coated FBG	−20	50	55.7	Present work

**Table 3 sensors-23-02811-t003:** Comparisons of the installation information for four ESG sensors and four FBG strain sensors that were mounted on the cantilever beam.

	ESG-S1/ESG-S2	FBG-S3/FBG-S4	ESG-S3/ESG-S4	FBG-S1/FBG-S2
L (mm)	307.0	257.0	228.0	164.0
W (mm)	31.0	25.9	22.9	16.6
Ratio (L/W)	9.903	9.923	9.956	9.879

## Data Availability

Data underlying the results presented in this paper are not publicly available at this time but may be obtained from the authors upon reasonable request.

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
