# Peer review of "Advantageous Strain Sensing Performances of FBG Strain Sensors Equipped with Planar UV-Curable Resin"

_sensors, 2023, doi:10.3390/s23052811_

Round 1

Reviewer 1 Report

See my notes in attached .pdf file 

Author Response

Response to Reviewer 1 Comments

Point 1:Based on my recommendations for authors (bullet points), the paper seem to be publishable after major corrections. The results are clearly presented - yes, methods are adequately described - yes... However, the most crucial missing point about this work is its missing novelty. Influence of lamination of fiber sensors into UV-curable resins has been richly described more than a decade ago. For example in: Smart Materials and Structures 20(12):125002 or in Measurement 45(9):2275 2280.

Response 1: Thanks for your questions. In our opinion, we agree that the work is not very novel. The existing FBG strain sensors have their limitations including having complex structure, a limited strain range and poor linearity performance, which limit their potential practical applications. However, in this paper, we pay more attention to the characteristics of the planar UV-curable resin coated FBG strain sensors with simple structure, a large strain range, excellent strain sensing performances including excellent repeatability and linearity [R2≥0.9998]. This is the major difference compared with previous work. After careful reading the Ref. [27-28] (i.e., Smart Materials and Structures 20(12):125002 and Measurement 45(9):2275 2280), the composite samples with polarimetric fiber or highly birefringent fiber sensors embedded in the multi-layer composite structure were fabricated. Although the resin is used both the above-mentioned references and our work, the fabricating processes and the research motivations of FBG sensors are different. In addition, the strain linearity of the polarimetric fiber sensors is limited (R2≤0.9992, as shown in Figure S1), which may limit their potential practical applications. This is probably because they use the fabric (very soft and have an uneven surface) as the substrate. However, in our work, the substrate is the stainless steel (is not soft and have a smooth surface), the strain linearity is excellent (R2≥0.9998, as shown in Figures 10-12).

Figure S1: (a) Comparison between characteristic of two side-hole sensors and (b) Strain response of two polarimetric sensors (side-hole and PCF). The data were extracted from previous work [28].

Changes: Lines: 60-63, we have added some descriptions to explain the differences between previous work and our work. In addition, we have added the two references [mentioned by the reviewer] in our revised manuscript.

Point 2:Language: language requires cross-reading by a native speaker, past tense is used instead of present tense, the sentences begin with “And” or “Which is”, confusing of basic physical terms “milimeters” instead of “widths” (line 138) or “temperature” instead of “heat” (line84), subjective interpretation of results (“Good optical properties…” -title of sec. 3.1 and3.2).

Response 2: The language has been revised and checked by a native speaker. The present tense has been used. The “And” or “Which is” have been changed or deleted in the sentences begin with “And” or “Which is”. The two basic physical terms have been changed. The “Good” has been deleted in the title of Sec. 3.1 and 3.2. In addition, the “Excellent” has been deleted in the title of Sec. 3.3.

Changes: We have revised the above-mentioned mistakes one by one and the revisions are marked in red in our revised manuscript.

Point 3:Methods: the paper sounds more like a student internship report than a scientific publication: authors tend to show too much of what they have done instead of focusing on major results and their interpretation. For example:

- photos of measurement of a width of something with a vernier caliper does not belong to a scientific paper (Figure 5). Instead, a graphical presentation of dependence of the shift of Bragg wavelength as a function of substrate thickness would be more appreciated here,

- description of the material mechanics methods in section 3.3 are more of a textbook style and can be shortened to an absolute required minimum.

Response 3: Thanks for your excellent suggestions. A graphical presentation of the relationship between the shift of Bragg wavelength and the coating thickness has been used in Figure 5. The description of the material mechanics methods in section 3.3 has been shortened.

Changes: We have changed the Figure 5 and we also have added some descriptions to describe Figure 5 (lines: 188-191). We have shortened the description of the material mechanics methods in section 3.3 (lines: 270-276).

Point 4:Figure 7, photo of the experimental setup is a pure chaos! It would be better to exchange it by a schematic of the setup.

Response 4: Thanks for your good suggestions. A schematic of experimental setup for strain measurements has been used in Figure 7.

Changes: We have changed the Figure 7.

Point 5:The physical interpretation of the measured data is a very weak point of this paper. For example, the sensitivity of FBG to temperature measurement (summarized in Table 2) has a very simple explanation concerning different coefficient of thermal expansion (CTE) of the fiber material (fused silica) and the coating material. Sensitivity of a pure fused silica sensor should be around 3pm/K at 1550nm. Since most of the polymer or metallic coating materials have CTE 10-100 times higher than glass, the interpretation of data from Table 2 is straight forward.

Response 5: The temperature sensitivity expression of the FBG embedded in the coating material has been listed, as shown in Equation (4) and the detailed descriptions have been added as well as.   In addition, according to Equation (2), we know that if the strain is equal to zero, the temperature sensitivity () of the bare FBG can be calculated as followed:

For the fused silica fiber,and[32, 33]. If,the temperature sensitivity is equal to 14.18 pm/℃.

In order to better compared with other FBG sensors, the maximal measured temperature sensitivity (i.e., 55.7 pm/℃) are used in Table 2. If the FBG embedded in the coating material, the temperature sensitivity of can be obtained according to the Equation (4).

Changes: We have added some descriptions to explain how to calculate the temperature sensitivity of the FBG embedded in the material coating, lines: 214-233. In addition, we have added some descriptions to explain why the SMSR of the FBG-S1 becomes worse and why the intensity of the FBG-S1 is reduced, lines:160-170. 

Point 6:Title: why is it underlined that the lamination of the sensor in the resin is “planar”? Does it have any consequence on the sensor performance? Would the results be the same when the lamination would be cylindrical, as it is the case in most of the fiber sensors? If there is no effect on the performance then the “planarity” of the lamination cannot be claimed as novelty issue.  

Response 6: There are some reasons that we choose the planar coating. Firstly, the planar coating can make sure that the distance between the FBG and the object (to be measured) strain sensor is very short (as shown in top view of Figure S2, i.e., d1<d2, because compared with d1, d2 includes the coating thickness and the cylindrical substrate thickness), resulting in a higher strain transfer efficiency when the FBG strain sensors are pasted on the surface of the object (to be measured). Secondly, compared with the cylindrical substrate, the thin planar substrate (0.06mm for our samples) is easily obtained, which is very convenient for us to fabricate the samples. Thirdly, compared with the thin planar substrate, the cylindrical substrate is not easily bended, which indicates that the FBG (embedded in cylindrical substrate) is not easily bended as well as, resulting in a large error between the measured value and the true value. Thus, in our work, the “planarity” of the lamination needs to highlight.

Figure S2: Schematic of (a) planar coating and (b) cylindrical coating for FBG sensors.

Changes: No change.

Point 7:The FBG spectra shown in Fig. 4 are quite broad (1nm) and irregular in shape. It is for me quite doubtful how a Bragg wavelength shift of 0.023nm (line 303) can be read out from such spectrum. 

Response 7: Actually, the FBG wavelength demodulator (T&S Communication Co, Ltd., TS-WI) with ultra-high resolution (the wavelength resolution is 1 pm, i.e., 0.001nm) is used in our experiments. Thus, the shift of 0.023 nm in wavelength can be easily read out from spectrum. The product specification of the FBG wavelength demodulator can be read from the following website:

Changes: We have added the necessary information of the FBG wavelength demodulator in our revised manuscript (lines: 252-253.)

Reviewer 2 Report

The authors carried out the detail investigations on the FBG strain sensors equipped with planar UV-curable resin. The good temperature sensitivity, the linearity of sensing results and the excellent repeatability were observed through the experimental test. The paper is well organized and written. I suggest to accept the manuscript after the minor revision for the following concerns:

1. What’s the major difference of the four FBGs under test? Any impact on the sensing performance due to the different structure? 

2. In Table 1, only the SMSR from the first FBG becomes worse, why? What’s the reason leads to the different results after the coating process? The optical intensity at the center wavelength also has the similar results (power reduction only observed in the first FBG), see in Fig. 4, Why?

Author Response

Response to Reviewer 2 Comments

The authors carried out the detail investigations on the FBG strain sensors equipped with planar UV-curable resin. The good temperature sensitivity, the linearity of sensing results and the excellent repeatability were observed through the experimental test. The paper is well organized and written. I suggest to accept the manuscript after the minor revision for the following concerns:

Point 1:What’s the major difference of the four FBGs under test? Any impact on the sensing performance due to the different structure?

Response 1: Thanks for your questions. The major difference of the four FBGs is that they have different coating thicknesses. The thicker coating thickness leads to the lager shift in wavelength (as shown in Figure 5) and the higher temperature sensitivity (as shown in Figure 6). According to the Equation (4), we can know that, the thicker coating thickness leads to a larger Cm, resulting in the higher temperature sensitivity. To facilitate understanding, the impact on the shift in wavelength and temperature sensitivity caused by the different coating thicknesses is listed as below:

FBG sensors (disorder)

Coating thickness (mm)

Shift in wavelength (nm)

temperature sensitivity (pm/℃)

FBG-S2

0.51

0.716

47.7

FBG-S1

0.54

0.932

52.7

FBG-S4

0.56

1.116

55.6

FBG-S3

0.63

1.264

55.7

Changes: We have changed the Figure 5 using a graphical presentation of the relationship between the shift of Bragg wavelength and the coating thickness. In addition, lines: 190-191, we have added some descriptions to highlight the major difference of the four FBGs. We also have added the temperature sensitivity expression of the FBG embedded in the coating material, as shown in Equation (4).

Point 2:In Table 1, only the SMSR from the first FBG becomes worse, why? What’s the reason leads to the different results after the coating process? The optical intensity at the center wavelength also has the similar results (power reduction only observed in the first FBG), see in Fig. 4, Why?

Response 2: The SMSR of the FBG-S1 becomes worse. According to previous work [29,30], it can attribute to the non-uniform strain distribution along the grating, because when the grating is subjected to the non-uniform strain, the intensity of side lobe (i.e., at 1538.919 nm in Fig. 4a) increases asymmetrically with the strain gradient increasing. According to previous work [23], the non-uniform strain is caused by the asymmetric axial force induced by the asymmetric coating, as shown in Figure S1. Thus, in order to avoid the asymmetric coating, before coating, we should put the FBG in the geometric middle of coating area. According to the expression of the reflection spectrum intensity [31], the larger FWHM (full width at half maximum) value leads to the smaller reflection intensity. Thus, the increase of the -3dB bandwidth is the reason why the intensity/power is reduced for the FBG-S1.

Changes: Lines: 160-171, we have added some descriptions to explain why the FBG-S1 have the worse SMSR and the reduced intensity after coating. In addition, we have also added some references in our revised manuscript.

Figure S1. (a) schematic of symmetric coating: the FBG is in the geometric middle of coating area. (b) schematic of asymmetric coating: the FBG is not in the geometric middle of coating area.

Round 2

Reviewer 1 Report

Please improve the quality of all figures, especially those including graphs. The numbers and letters included in the graphs are not sharp and hardly readable.

Author Response

Thank you for your good suggestions. We have improved the quality of all Figures and Tables, and the numbers and letters included in the graphs are more clear and readable